# Reducing Training Sample Memorization in GANs by Training with Memorization Rejection

## Abstract

Generative adversarial network (GAN) continues to be a popular research direction due to its high generation quality. It is observed that many state-of-the-art GANs generate samples that are more similar to the training set than a holdout testing set from the same distribution, hinting some training samples are implicitly memorized in these models. This memorization behavior is unfavorable in many applications that demand the generated samples to be sufficiently distinct from known samples. Nevertheless, it is unclear whether it is possible to reduce memorization without compromising the generation quality. In this paper, we propose memorization rejection, a training scheme that rejects generated samples that are near-duplicates of training samples during training. Our scheme is simple, generic and can be directly applied to any GAN architecture. Experiments on multiple datasets and GAN models validate that memorization rejection effectively reduces training sample memorization, and in many cases does not sacrifice the generation quality.

## 1 Introduction

There has been much progress made on improving the generation quality of Generative Adversarial Networks (GANs) (Brock et al., 2019; Goodfellow et al., 2014; Karras et al., 2020; Wu et al., 2019; Zhao et al., 2020; Zhang et al., 2019). Despite GANs being capable of generating high-fidelity samples, it has been recently observed that they tend to memorize training samples due to the high model complexity coupled with a finite amount of training samples (Meehan et al., 2020; Lopez-Paz & Oquab, 2018; Gulrajani et al., 2020; Borji, 2021). This naturally leads to the following questions: Are GANs learning the underlying distribution or merely memorizing training samples? More fundamentally, what is the relationship between learning and memorizing for GANs? Studying these questions are important since generative models that output near-duplicates of the training data are undesirable for many applications. For example, Repecka et al. (2021) proposed to learn the diversity of natural protein sequencing with GANs and generate new protein structures to aid medicine development. Frid-Adar et al. (2018) leveraged GANs to generate augmented medical images and increase the size of training data for improving liver lesion classification.

Although measuring and preventing memorization in supervised learning is well-studied, handling memorization in generative modeling is non-trivial. For supervised learning, training sample memorization typically results in overfitting and can be diagnosed by benchmarking on a holdout testing dataset. In contrast, a generative model that completely memorizes the training data and *only* generates near-duplicates of the training data can still perform well on common distribution-matching-based quality metrics, even when evaluated on a holdout testing set.

Recently various metrics and detection methods have been proposed to analyze the severity of memorization after GAN models are trained (Borji, 2021; Bounliphone et al., 2016; Esteban et al., 2017; Lopez-Paz & Oquab, 2018; Liu et al., 2017; Gulrajani et al., 2020; Thanh-Tung & Tran, 2020; Nalisnick et al., 2019). Some of these methods rely on training a new neural network for measuring sample distance while others rely on traditional statistical tests. However, it is still unclear how to actively reduce memorization during GAN training. We thus aim to answer the following questions in this paper: is it possible to efficiently

reduce memorization during the training phase? If so, to what extent can memorization be reduced without sacrificing the generation quality? Our contributions are as follows:

1. We confirmed that while the distance of a generated instance to the training data is generally correlated with its quality, it is not the case for instances that are already sufficiently close. Therefore, it is possible to reduce memorization without sacrificing generation quality.

2. We propose memorization rejection, a simple training scheme that can effectively reduce memorization in GAN. The method is based on the key insight that a generated sample being sufficiently similar to its nearest neighbor in the training data implies good enough quality and further optimizing it causes the model to overfit and memorize. To the best of our knowledge, this is the first method proposed for reducing training data memorization in GAN training.

3. Experimental results demonstrate that our proposed method is effective in reducing training sample memorization. We provided a guideline for estimating the optimal hyperparameter that maximally reduces memorization while minimally impacting the generation quality.

## 2 Preliminaries

Consider an input space $\mathcal{X}$ and an $N$-dimensional code space $\mathcal{Z} = \mathbb{R}^N$. For instance, when considering RGB images, $\mathcal{X}$ is simply $\mathbb{R}^{3 \times w \times h}$, where $w$ and $h$ are respectively the width and height of the image (in this paper, $\mathcal{X} = \mathbb{R}^{3 \times w \times h}$ if not specified otherwise). Generative adversarial networks (GANs; Goodfellow et al., 2014) typically consist of a generator function and a discriminator function. The generator function $G_\theta \colon \mathcal{Z} \to \mathcal{X}$, parameterized by $\theta \in \Theta$, decodes from $\mathcal{Z}$ to $\mathcal{X}$. The discriminator function $D_\phi \colon \mathcal{X} \to \mathbb{R}$, parameterized by $\phi \in \Phi$, maps any $x \in \mathcal{X}$ to a real value that reflects how likely $x$ comes from an underlying distribution $p(\mathcal{X})$. A typical objective of a GAN optimizes the minimax loss between $G_\theta$ and $D_\phi$

$$\min_{\theta \in \Theta} \max_{\phi \in \Phi} \mathop{\mathbb{E}}_{x \sim p(\mathcal{X})} [\log D_\phi(x)] + \mathop{\mathbb{E}}_{z \sim q(\mathcal{Z})} [\log(1 - D_\phi(G_\theta(z)))],$$

where $q(\mathcal{Z})$ is a controllable distribution (e.g. Gaussian). GANs aim to approximate $p(\mathcal{X})$ by $G_\theta(q(\mathcal{Z}))$ with the adversarial help of the discriminator $D_\phi(x)$. In particular, the generator is optimized to increase the likelihood of generated instances with the likelihood gauged by the discriminator, while the discriminator is optimized to increase the likelihood of instances sampled from the real distribution $p(\mathcal{X})$ and decrease the likelihood of instances generated from the fake distribution $G_\theta(q(\mathcal{Z}))$. Since it is infeasible to sample from $p(\mathcal{X})$ directly, a training set $X_T \subseteq \mathcal{X}$ of $N$ instances is used to approximate the population instead.

### 2.1 Quantitative evaluation of sample similarity

The most commonly used method to detect training sample memorization is by visualizing nearest neighbors of generated images in the training data (Brock et al., 2019; Borji, 2018). If the visualized samples look similar to their nearest neighbors in the training data, it is reasonable to suspect that the model is trying to memorize the training data. Given a generated sample $x \sim G_\theta(q(\mathcal{Z}))$ and an embedding function $f : \mathcal{X} \to \mathbb{R}^k$, the nearest neighbor of $x$ in the training set is defined as

$$\mathrm{NN}_{f, X_T}(x) = \argmin_{x' \in X_T} 1 - \frac{\langle f(x), f(x') \rangle}{\|f(x)\| \cdot \|f(x')\|}.$$

The cosine similarity is conventionally used for evaluating the similarity of latent vectors (Salton & Buckley, 1988; Le-Khac et al., 2020; Borji, 2021) but other distance metrics could also be chosen. To avoid sensitivity to noise in the input space, $f$ is usually chosen to project to a latent space embedded with higher-level semantics. It is widely believed that a pretrained image classification model can extract high-level semantics and serves as a robust latent space for distance measurement. For example, calculation of FID involves first passing the set of images through the Inception v3 (Szegedy et al., 2016) classification model pretrained on ImageNet for feature extraction. A well-chosen $f$ retrieves nearest neighbors that align well with human's

perception. Following this definition, the distance to the nearest neighbor can serve as a quantitative measure for sample similarity

$$d_{f,X_T}(x) = \min_{x' \in X_T} 1 - \frac{\langle f(x), f(x') \rangle}{\|f(x)\| \cdot \|f(x')\|}.$$

Thus, the problem of reducing memorization can be formulated as regulating the nearest neighbor distance of generated samples, which motivates our proposed algorithm.

## 2.2 Quantitative evaluation of memorization

Meehan et al. (2020) proposed a non-parametric test score $C_T$ for measuring the degree of training sample memorization of a generative model based on sample similarity. Their key insight is that a model should generate samples that are on average, as similar to the training data as an independently drawn test sample from the same distribution. The model is memorizing if the generated samples are on average, *more* similar to the training data than an independently drawn test sample from the same distribution.

The memorization test is based on the Mann-Whitney U test, a non-parametric statistical test for testing the ordinal relationship with the null hypothesis that the given two sets of samples are from the same distribution. In this case, the two sets of samples are the nearest neighbor distances (with respect to the training data) of a generated set and a reference testing set. The more severe the memorization, the more negative the U statistics, and vice versa. Additionally, to better detect local memorization, the input domain can be divided into subspaces and the test score is aggregated over memorization tests performed on each of the subspaces. In this paper, we adopt the definition of memorization as characterized by the $C_T$ values.

## 2.3 Generation quality and memorization

Good generation quality and reduced memorization can coexist. In the ideal case, if the generator perfectly fits the underlying data distribution, then the generated samples have perfect quality and are in no way more similar to the training data than another independent sample from the distribution. However, GAN models are imperfect. Figure 1 shows the nearest neighbor distance distribution (approximated by 2K samples) of a generated set from BigGAN and a reference testing set (CIFAR10.1). If the model successfully learned the data distribution, the expectations of the two nearest neighbor distributions should be identical. However, samples generated from BigGAN (orange line) are in fact closer to the training data than samples from the reference testing set (highlighted in orange) which indicates the memorization phenomenon.

In general, it is true that generated samples with smaller nearest neighbor distances are associated with better quality. Smaller distances imply being closer to the training distribution. Figure 2 visualizes a subset of 5k samples from a BigGAN trained on CIFAR10. The images are sorted by their nearest neighbor distance. From top to bottom, each row shows 10 images from the 20%, 40%, 60%, 80%, and 100% percentile, respectively. The upper rows with lower nearest neighbor distance are associated with better perceptual quality. This confirms nearest neighbor distance in general is an indicator of quality.

However, the nearest neighbor distance is *not* an indicator of quality when the distance is sufficiently small. If a generated sample is already close to the data distribution, a smaller nearest neighbor distance only implies higher similarity with the training sample. Figure 3 visualizes a subset of 5k samples from a BigGAN trained on CIFAR10, sorted by their nearest neighbor distance. From top to bottom, each

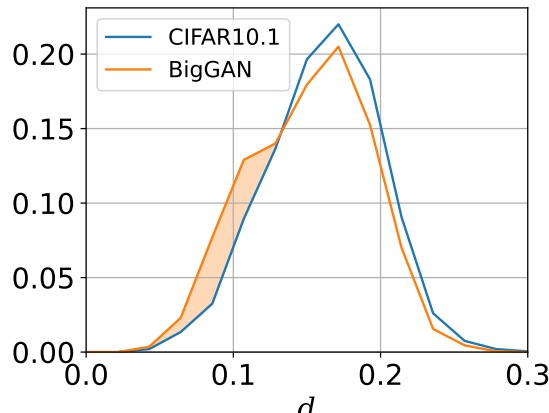

Figure 1: Nearest neighbor distance distribution of the reference testing set (CIFAR10.1) versus BigGAN.

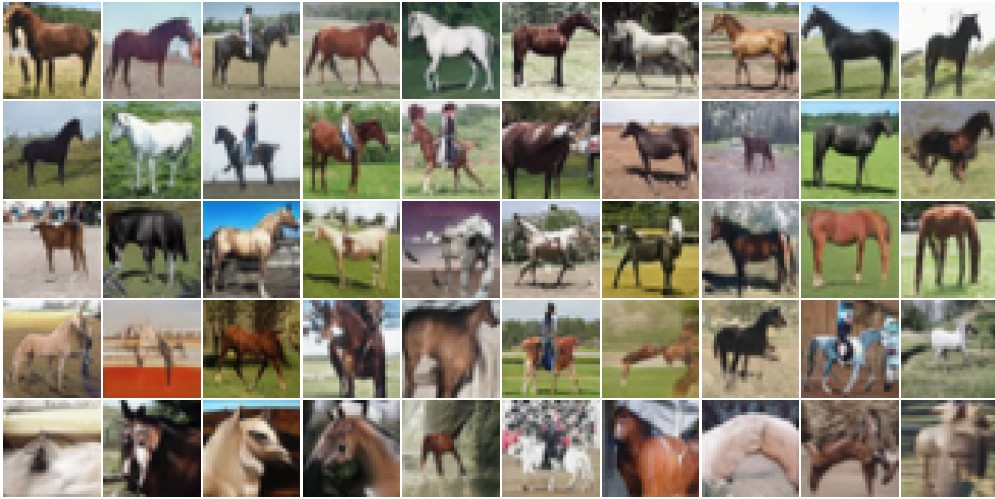

Figure 2: Visualize CIFAR10 "horse" samples from BigGAN, sorted with the nearest neighbor distance. From top to bottom each row shows the 20%, 40%, 60%, 80%, and 100% percentile.

row shows 10 images from the 4%, 8%, 12%, 16%, and 20% percentile, respectively. There is no perceptible quality difference between rows. Thus, for generated samples already sufficiently close to the training data, their nearest neighbor distances are indicative of potential memorization, not quality.

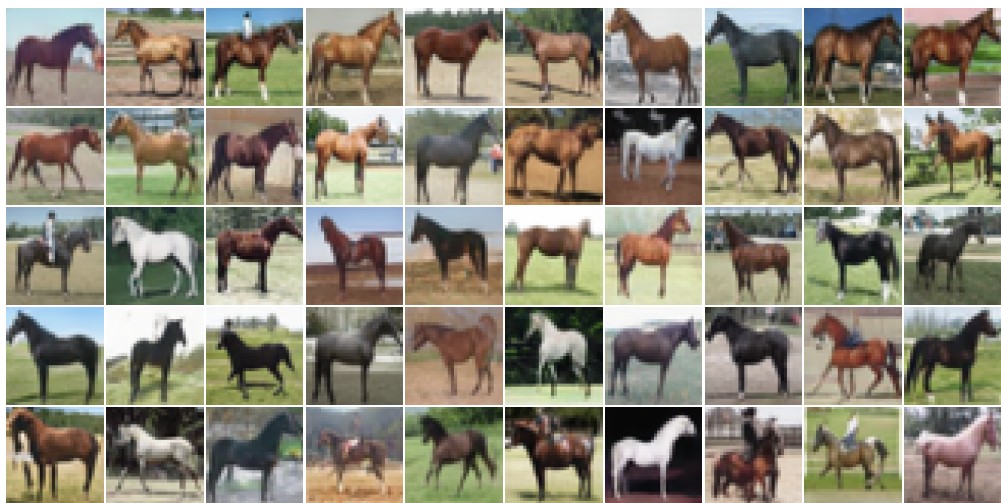

Figure 3: Visualize CIFAR10 "horse" samples from BigGAN, sorted with the nearest neighbor distance. From top to bottom each row shows the 4%, 8%, 12%, 16%, and 20% percentile.

The issue for learning the distribution with GANs is only having access to a finite number of training data. The data distribution is approximated by a joint Dirac delta distribution of training samples. As training progresses, the generated samples become more similar to the training data (see Figure 4).

Coupled with model over-parametrization, the learned implicit likelihood is overly high for neighborhoods near training samples. Yang & E (2021) proved that the distribution learned with GANs either diverges or converges weakly to the empirical training data distribution. The authors proved that early stopping allows quality measured by Wasserstein metric to escape from the curse of dimensionality, despite inevitable memorization in the long run.

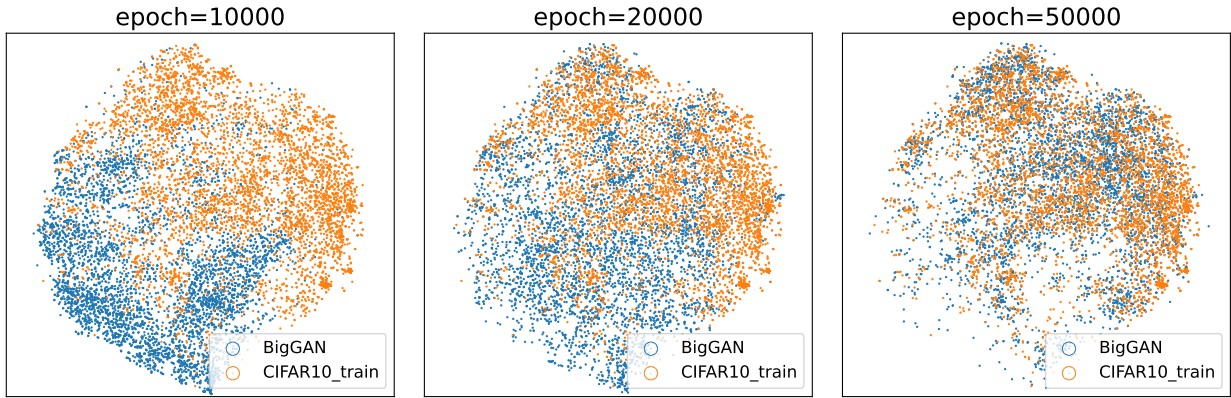

Figure 4: tSNE plot of CIFAR10 "car" training and generated samples from different stages of training BigGAN

Typically the training of GANs is terminated when the quality metric starts to deteriorate (e.g. FID starts to increase). However, Meehan et al. (2020) observed memorization in state-of-the-art GAN models, implying deterioration of quality metrics is not a sufficient criteria to prevent memorization. It is also unreasonable to expect the entire distribution to be learned at equal speed. Some easier parts of the distribution might already be well-fitted and starting to memorize while other more difficult parts of the distribution require more epochs to learn. For example, in conditional generation tasks such as CIFAR10 (Krizhevsky et al., 2009) the difficulty of learning each class is different. It is much easier to learn manmade objects with more clearly defined borders (e.g. cars, trucks, and ships) than animals with similar color as the background (e.g. deer, birds, and frogs). Thus, early stopping a model potentially leads to both underfitting and overfitting (memorization) for different parts of the distribution.

## 3 Methods

Recall that the difference in nearest neighbor distance distributions shown in Figure 1 reflects memorization in GAN. To reduce memorization, the likelihood of the orange region should be reduced. We propose a simple and effective method to achieve this goal.

### 3.1 Memorization rejection

We want to regularize the model to avoid generating samples overly similar to the training data. As shown in Section 2.3, samples that are already close to the training data have good-enough quality. Pushing them further towards the training data results in memorization instead of quality improvement. Based on the premise, we proposed **Memorization Rejection (MR)** which rejects generated samples that resemble near duplicates of the training data (see Algorithm 3.1). This is achieved by setting a predefined rejection threshold $\tau$ on the nearest neighbor distance. The new objective is modified as follows

$$\min_{\theta \in \Theta} \max_{\phi \in \Phi} \mathop{\mathbb{E}}_{x \sim X_T} \left[ \log D_\phi(x) \right] + \mathop{\mathbb{E}}_{\hat{x} \sim G_\theta(q'(\mathcal{Z}))} \left[ \log(1 - D_\phi(\hat{x})) \right]$$

$$q'(z) := \begin{cases} \frac{q(z)}{Q}, & \text{if } d_{f,X_T}(G_\theta(z)) \geq \tau \\ 0, & \text{otherwise} \end{cases}$$

where $Q$ is the normalizing constant. To understand the effect of memorization rejection, we can rewrite the the latter component of the objective

$$\mathop{\mathbb{E}}_{\hat{x} \sim G_\theta(q'(\mathcal{Z}))} \left[ \log(1 - D_\phi(\hat{x})) \right] = \mathop{\mathbb{E}}_{\hat{x} \sim G_\theta(q(\mathcal{Z}))} \left[ \frac{\log(1 - D_\phi(\hat{x}))}{Q} \right] + l_r$$

$$l_r = -\int_z \log(1 - D_\phi(G_\theta(z))) \cdot q''(z)\mathrm{d}z$$

$$q''(z) := \begin{cases} \frac{q(z)}{Q}, & \text{if } d_{f,X_T}(G_\theta(z)) < \tau \\ 0, & \text{otherwise} \end{cases}$$

The new objective becomes the original GAN minimax loss plus the regularization term $l_r$ which penalizes $G_\theta$ for generating samples with nearest neighbor distance less than $\tau$, i.e., overly similar to the training data. Memorization rejection can be viewed as a form of adaptive early stopping. The training stops for generated samples with sufficiently good quality (nearest neighbor distance less than $\tau$) while other samples continue to be updated and improved.

Memorization rejection is performed when updating the generator and discriminator according to the objective. However, in practice performing MR only when training the generator is sufficient. We suspect the discriminator requires all the generated (fake) samples to accurately estimate the likelihood, which is related to how discriminators are updated multiple times before updating the generator once (Arjovsky et al., 2017). A partially converged discriminator can provide better feedback to the generator for generating realistic samples. The method is effective as long as the generator is penalized for memorization.

Note that rejection is only performed during training. For testing, samples are drawn from the original distribution $\hat{x} \sim G_\theta(q(\mathcal{Z}))$ as MR is effectively a regularization term $l_r$ and should not be applied during evaluation. The goal is still to learn the mapping from latent codes $q(\mathcal{Z})$ to the real data distribution $p(X)$.

Training GAN with Memorization Rejection [1] Rejection Sampling$\tau, G_\theta, d(\cdot), q$ $d \leftarrow 0$ $d \leq \tau$ Sample $z$ from $q(\mathcal{Z})$ $\hat{x} \leftarrow G_\theta(z)$ $d \leftarrow d(\hat{x})$ **return** $\hat{x}$ Training with MR$X_T, G_\theta, D_\phi, N, \tau, d(\cdot), q$ $i = 1, \ldots, N$ Sample $x$ uniformly from $X_T$ Sample $z$ from $q(\mathcal{Z})$ $\hat{x} \leftarrow G_\theta(z)$ Update $D_\phi$ with $x$ and $\hat{x}$

$\hat{x} \leftarrow$ Rejection Sampling$(\tau, G_\theta, d(\cdot), q)$ Update $G_\theta$ with $\hat{x}$ return $G_\theta$ and $D_\phi$

### 3.2 Computational complexity

Performing memorization rejection requires projecting samples to the latent space and calculating the nearest neighbor distance in each generator update. For each generator update, the additional forward pass through the embedding function $f$ and the distance calculation result in twice the total amount of training time in our experiments. We conducted exact nearest neighbor search in all our experiments as the overhead is tolerable (less than 2x). To further speed up training on larger scale datasets, approximated nearest neighbor search (Li et al., 2020; Malkov & Yashunin, 2018) can be applied instead. Open source libraries (Guo et al., 2020; Jayaram Subramanya et al., 2019) allow efficient approximated nearest neighbor search on billion-scale datasets.

## 4 Related work

### 4.1 GAN memorization metrics

Many works have studied different definitions of memorization in GANs and some of which proposed metrics to quantify the severity of memorization (Meehan et al., 2020; Lopez-Paz & Oquab, 2018; Bounliphone et al., 2016; Gulrajani et al., 2020; Borji, 2021; Webster et al., 2019; Adlam et al., 2019; Feng et al., 2021; Bai et al., 2021). There is a line of studies that relies on sample-based statistical tests. Lopez-Paz & Oquab (2018) applied the Two-Sample Nearest Neighbor non-parametric test to evaluate the leave one out accuracy of the nearest neighbor classifier evaluated on a dataset consisting of generated samples and samples from the original training set. Esteban et al. (2017) adopted the result of maximum mean discrepancy three sample test (Bounliphone et al., 2016) as the null hypothesis and evaluated the averaged p-values. Meehan et al. (2020) proposed a non-parametric test which estimates the likelihood of the nearest neighbor distance of a generated sample being greater than a sample from a reference test set.

Another line of studies relies on the Neural Network Divergence for measuring the overall generalization of generative models (Liu et al., 2017; Gulrajani et al., 2020). The method requires training a neural network

to differentiate samples from two distributions and using the converged loss after training as a proxy for the discriminability of the two distributions. On a tangential perspective, Webster et al. (2019) measures memorization by retrieving latent code that maps to near duplicates of training samples.

## 4.2 Data augmentation in GAN training

Data augmentation have been applied to reduce overfitting in GANs (Zhao et al., 2020; Karras et al., 2020; Tran et al., 2021; Liu et al., 2021; Tseng et al., 2021; Yang et al., 2021). However, the overfitting that can be solved by data augmentation refers to overfitting of the discriminator when only a limited amount of training data is available. The augmented training data improves generation quality by preventing the discriminator from making high confidence predictions. However, studies on GAN data augmentation techniques are not shown to reduce generator training sample memorization, as defined in this work. In fact, we later show in section 5.4 that data augmentation (Zhao et al., 2020) is ineffective at reducing memorization.

## 4.3 Sampling with rejection in GANs

Sampling is essential to training GANs since the GAN objective is based on the two-sample test of real and fake distributions. One common technique to efficiently sample from a complex distribution is rejection sampling. Rejection sampling can be further generalized to "sampling with rejection", where the criteria for rejecting a sample depends on a custom function as opposed to the probability density. This allows straightforward filtering of unfavorable samples.

The criteria for rejection can be customized for different needs. Lim & Ye (2017) proposed to adopt hinge loss as the objective and rejects a sample (from being used to update the model) if it falls within the margin. Azadi et al. (2018) proposed a post-processing scheme where generated samples are rejected based on the likelihood estimated by the discriminator. Their key insight is it is easier for the discriminator to determine when the distribution is *not* being modeled precisely. Sinha et al. (2020) rejects samples associated with lower likelihood estimated by the discriminator during training and only update the generator with "good quality samples" to improve generation quality.

## 5 Experimental Results

Our goal is to analyze how training GANs with memorization rejection affects the performance in terms of generation quality and memorization severity. We demonstrate that it is possible to reduce memorization with minimal (non-perceivable) impact on the generation quality. The code for all the experiments will be open sourced.

### 5.1 Experimental setting

We trained GAN models with different rejection thresholds $\tau$. Higher rejection thresholds imply generated samples must be more distinct from the training data to be used for updating the model. The models are benchmarked with FID for generation quality and $C_T$ score defined in Section 2.2 for memorization severity. Each experiment is repeated 4 times for consistency and the average performance is reported. A latent projection function $f$ is required to calculate FID and $C_T$. We chose the same projection $f$ for both metrics, allowing the evaluation of quality and memorization severity be in the same latent space. Following the convention for FID (Heusel et al., 2018), we chose the penultimate layer of the Inception v3 model (Szegedy et al., 2016) pretrained on ImageNet as the embedding function $f$.

#### 5.1.1 Datasets and models

We conducted conditional generation experiments on the CIFAR10 dataset (Krizhevsky et al., 2009). The dataset consists of 50k training samples of natural images in 10 classes. We experimented on models of different complexity: SAGAN (Zhang et al., 2019), BigGAN (Brock et al., 2019), and BigGAN with differential augmentation (Zhao et al., 2020). The three models are intentionally selected to be incrementally more complex. BigGAN with differential augmentation is built upon BigGAN while BigGAN adopts the

self-attention mechanism in SAGAN. We expect that higher complexity causes models to memorize the training data more. The training hyperparameters are set as follows (no additional finetuning):

- Batch size: 64.

- Total number of steps for training: 100,000.

- Learning rate: 0.0002 (for the generator and discriminator).

- Optimized with ADAM ($\beta_1$: 0.5, $\beta_2$: 0.999).

- The discriminator is updated for 5 steps per one step update on the generator.

### 5.1.2 Reference testing set

It is important to obtain an unbiased memorization measurement by using an independent test set disjoint from the training set. The more distinct and independent the reference testing set is, the more accurate the evaluation of memorization severity. Although the CIFAR10 testing set is the most accessible choice, Barz & Denzler (2020) identified the issue of high overlap between training and testing set of CIFAR10. They mined duplicate images in the testing set using the projected cosine similarity as the nearest neighbor distance (measured with respect to the training set) and manually replaced near duplicates to construct a new dataset ciFAIR10. Unfortunately upon closer inspection, many near duplicates still exist in the ciFAIR10 dataset, possibly due to only considering the 1-nearest neighbor when mining.

Recht et al. (2018) constructed CIFAR10.1 to serve as a new benchmark dataset for CIFAR10 to verify whether state-of-the-art classification models can generalize to new samples. They resampled from the Tiny Images repository and went through the exact process of creating the CIFAR10 dataset with additional rigorous data-cleaning procedures. This includes manually inspecting the 10 nearest neighbors and removing instances of near duplicates. They observed that good model performance on the existing CIFAR10 testing set don't necessarily transfer to CIFAR10.1 and suggested important modes missing from the existing testing set. Thus, we selected CIFAR10.1 (Recht et al., 2018) as the reference testing set for metric evaluation since it is curated to have the least overlap with the training data.

## 5.2 Effectiveness of memorization rejection

Recall we showed in Figure 1 that samples generated from GANs are distributed closer to the training data than a reference testing set. In figure 5, in addition to the BigGAN distribution and the reference testing set (CIFAR10.1), we further plot the distribution generated by BigGAN with memorization rejection. We observe that the distribution of BigGAN with MR is more similar to the reference testing set, indicating training with memorization rejection reduces the similarity to the training data. This provides qualitative evidence that memorization rejection is effective in reducing training sample memorization.

Next we analyze the generation quality and memorization severity quantitatively. Figure 6 shows the FID and $C_T$ of BigGANs trained with different rejection thresholds. Recall $C_T$ value of 0 implies the generated samples are as similar to the training data as the reference testing set, i.e., no memorization. Negative $C_T$ values implies memorization. We observe for rejection thresholds up to 0.13, $C_T$ decreases with minimal impact on the FID. The memorization is reduced without degrading generation quality within this region. Figure 7 visualize generated samples from BigGAN trained with no rejection and with $\tau = 0.13$. There is no visually perceptible generation quality different between the two sets of samples.

For thresholds greater than 0.13, the reduction in memorization comes with tradeoff in quality. For $\tau = 0.16$ the $C_T$ remains negative, indicating slight memorization. Although training sample memorization is not completely removed (which may not even be possible), we demonstrated that training GANs with memorization rejection reduces the severity. The rejection threshold serves as a control knob to regularize the model and in this experiment a properly tuned threshold of $\tau = 0.13$ improves $C_T$ while maintaining (even improving) FID.

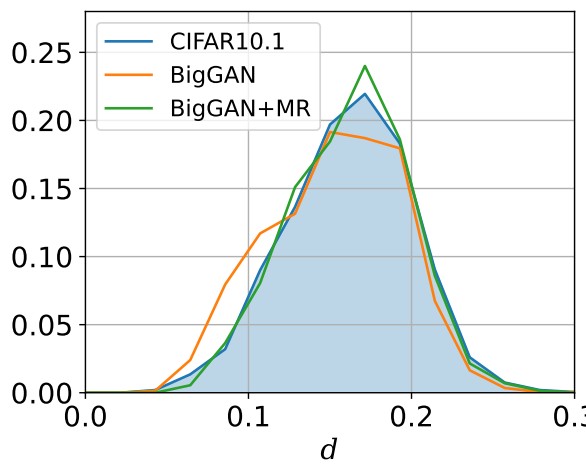

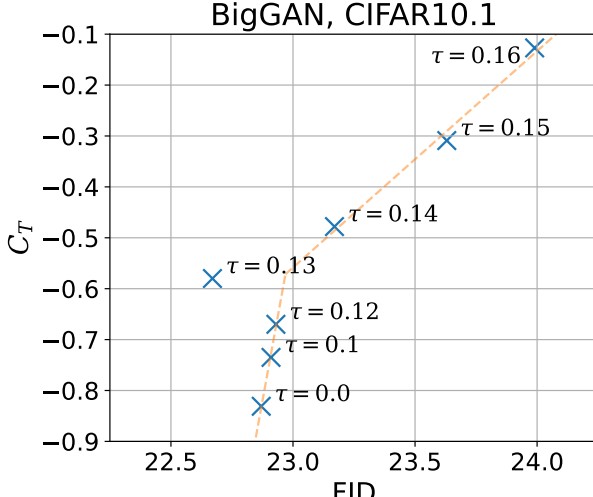

Figure 5: Nearest neighbor distance distribution (CIFAR10 train) of the reference testing sets.

Figure 6: Relation between quality (FID) and memorization ($C_T$) with varying rejection threshold $\tau$.

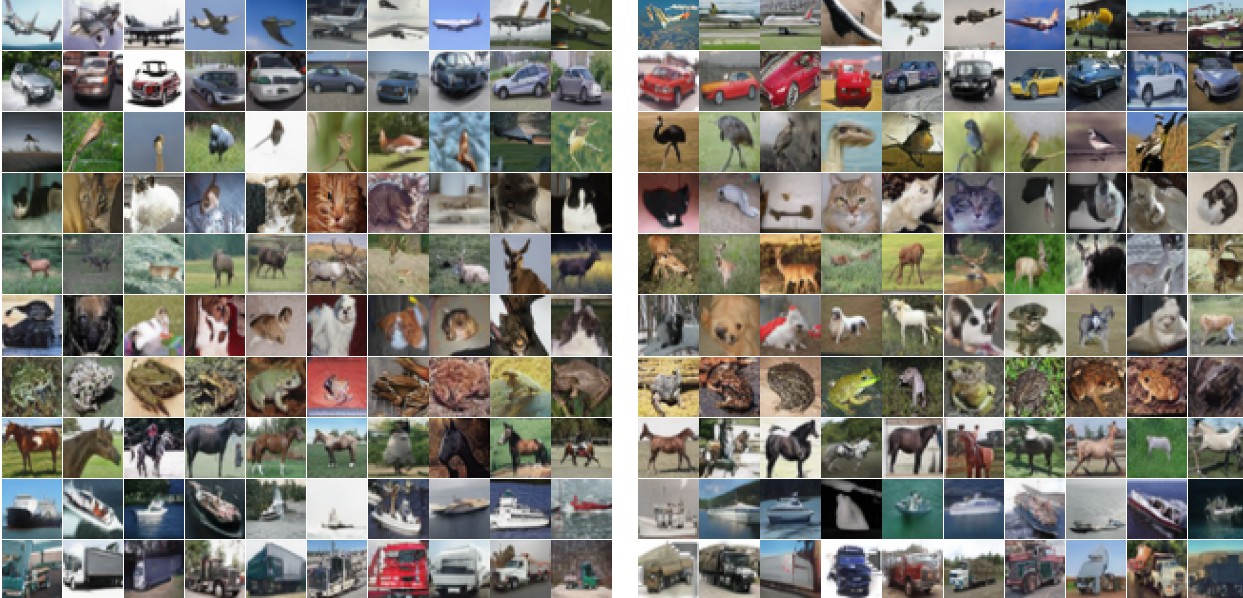

Figure 7: Visualize non-curated, generated samples. Left: BigGAN trained *without* MR. Right: BigGAN trained *with* MR. The generation quality of the model with and without MR is similar.

## 5.3 Class-wise memorization severity

One incentive to adopt memorization rejection is to serve as a form of adaptive early stopping to allow regions in the distribution to be learned with different speeds. For conditional generation, the difficulty of learning each class is different. As mentioned in Section 2.3, manmade object classes (e.g. car, ship, plane) are easier than natural object classes (e.g. bird, cat, frog) to learn. Figure 8 shows the classwise $C_T$ values for models trained with different rejection thresholds. As a sanity check, $C_T$ values indeed reflect how easily a class is learned. Coincidentally, classes that are easier to learn are also more easily memorized. MR reduces the memorization severity of highly memorized classes.

Besides reducing memorization of memorized classes, one thing to be aware of is whether memorization rejection causes underfitting classes to be even more underfitting. It is not ideal if memorization rejection shifts the entire generated distribution uniformly away from the training data. The target for memorization rejection is the highly memorized regions only. According to our experiments, MR is effective for classes with more severe memorization, i.e., more negative $C_T$ values. On the other hand, classes that are not memorized, i.e., more positive $C_T$ values are barely affected.

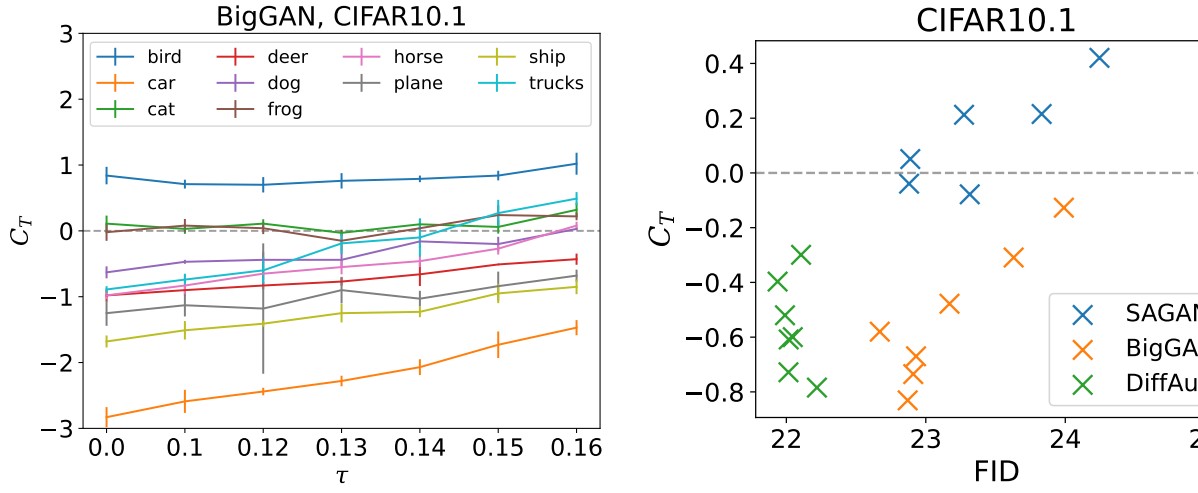

Figure 8: Effect of rejection threshold $\tau$ on memorization ($C_T$) of each CIFAR10 class.

Figure 9: Relation between quality (FID) and memorization ($C_T$) with varying $\tau$ for different models.

### 5.4 Experiments on model architectures

Figure 9 shows that not only is memorization rejection generic and applicable to any GAN model, it is also effective. The complexity of the models from lowest to highest is SAGAN, BigGAN, and BigGAN with differential augmentation. The complexity is reflected on the performance of both metrics, where more complex models is associated with lower FID scores (better quality) and more negative $C_T$ values (more memorization). We observe that for all three models, there exists some rejection threshold that improves the $C_T$ value without compromising the quality.

Differential augmentation (Zhao et al., 2020) is a technique for augmenting training data to avoid the discriminator overfitting while preventing the generator from learning the augmented samples. Figure 9 shows that BigGAN with differential augmentation exhibits the most severe memorization when no MR is applied. This indicates that augmentation techniques in GAN training, albeit useful for improving the generation quality, is not effective against reducing training sample memorization. However, the $C_T$ values can be significantly improved by increasing the rejection threshold without any degradation in quality as measured with FID when coupled with MR. This showcases the compatibility of memorization rejection with other tangential GAN training techniques.

### 5.5 Evaluation with various distance metrics

To perform memorization rejection, a distance metric $d_{f,X_T}$ is required to evaluate the nearest neighbor distance during training. Although the GAN model does not directly optimize for the distance, the rejection can be viewed as a regularization term on the original GAN objective as derived in section 3.1. Thus, there is a risk of the generator adversarially learning to generate samples that are dissimilar to the training data gauged with the distance metric used during training, but does not lead to actual reduction in memorization. If the memorization reduction is indeed legitimate, we should observe the same improvement in $C_T$ when evaluation with other distance metrics.

We constructed different distance metrics by changing the embedding function $f$. Specifically, we selected the penultimate layer of different ImageNet pretrained models as the embedding space, which are expected to be rich in high-level semantics but not equivalent. Figure 10 shows the $C_T$ values evaluated with different distance metrics. The universal trend holds across all the metrics. Higher rejection thresholds yield less negative $C_T$ values. This provides quantitative evidence that the observed improvement in $C_T$ is not merely an artifact of MR, but represents actual reduction in memorization severity.

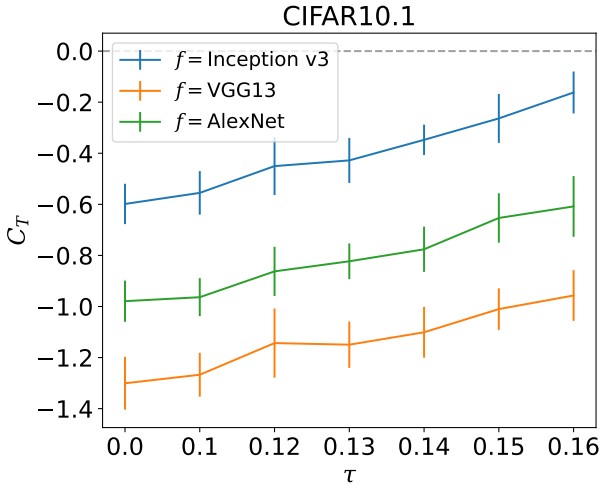

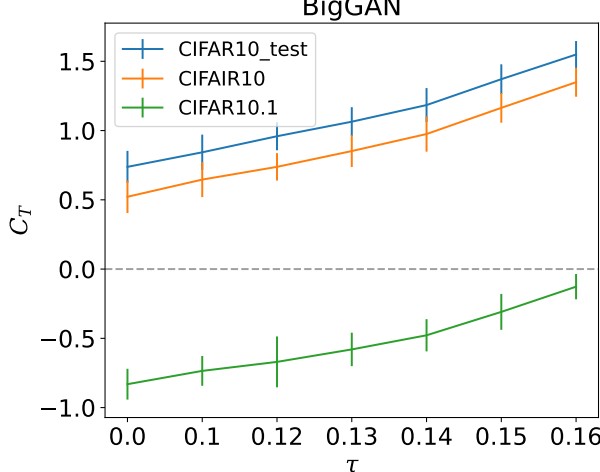

Figure 10: Effect of rejection threshold $\tau$ on memorization ($C_T$) evaluated with different distance function $d_{f,X_T}$. The result suggests applying memorization rejection *does not* cause the model to adversarially optimize for the distance function applied during training. Rather it effectively reduces memorization.

Figure 11: Effect of rejection threshold $\tau$ on memorization ($C_T$) evaluated with respect to different testing sets. The positive correlation between the rejection threshold and increased $C_T$ value is consistent across the testing sets.

## 5.6 Evaluation with various reference testing sets

Recall that we emphasized the importance of selecting a proper reference testing set in section 5.1.2. The distinctiveness of the reference testing set from the training data reflects the strictness of the criteria for memorization. Our choice of CIFAR10.1 (Recht et al., 2018) for reference in our experiments indicates a higher standard for the models. What happens if a testing set with more overlap with the training data is chosen as reference instead?

Figure 11 shows the $C_T$ values when using other testing sets as reference. CIFAR10 test is the testing set included in the original CIFAR10 dataset, which is found to be highly overlapping with the training set (Recht et al., 2018). CIFAIR10 is the testing dataset curated by Barz & Denzler (2020) as an attempt to remove some near duplicates of the training data but not all of them. The testing sets ranked from the degree of overlap with the training data from high to low would be CIFAR10 testing, CIFAIR10, then CIFAR10.1. The absolute $C_T$ values of the testing sets reflect their distinctiveness from the training data. The less distinctive the higher the $C_T$ value, and the lower the criteria for training sample memorization. On the other hand, if we compare the relative $C_T$ values within the same testing set, the positive correlation between the rejection threshold and $C_T$ value holds. This suggests that even if a non-overlapping, independent reference isn't available, the effectiveness of MR can still be observed.

## 5.7 Guideline for threshold selection

We have shown in previous sections that a well-chosen rejection threshold allows reduced memorization with minimal impact on generation quality. Due to the tradeoff nature between quality and memorization illustrated in section 2.3, we showed through experiments that the optimal rejection thresholds can and

should be tuned. It is only natural to ask: is it possible to estimate the neighborhood of where the optimal threshold lies without brute-force trial and error?

The initial intention for performing memorization rejection is to avoid updating generated samples that are too similar to their training data nearest neighbors. How the data is distributed in the latent space for distance evaluate is key to determining the rejection threshold. We can estimate the average density of the data distribution by measuring the average nearest neighbor distance of the training data

$$\bar{d}_{X_T} = \frac{1}{|X_T|} \sum_{x \in X_T} d_{f,X_T \setminus \{x\}}(x).$$

An generated sample with nearest neighbor distance less than $\bar{d}_{X_T}$ is likely close to the data distribution which implies "good enough" quality. This makes $\bar{d}_{X_T}$ a natural initial choice for performing memorization rejection.

The average nearest neighbor distance $\bar{d}_{X_T}$ of the CIFAR10 training set with Inception v3 model pretrained on ImageNet as the embedding function $f$ is 0.15. Figure 12 shows the tSNE figures for CIFAR10 generated samples partitioned according to their nearest neighbor distance $d_{f,X_T}$. Generated samples with $d$ less than 0.15 covers most of the data distribution while samples with $d$ greater than 0.15 fall outside the data distribution. This suggest the average nearest neighbor distance of the training samples can be used to determine whether a generated sample is close enough to the data distribution and serve as the rejection threshold.

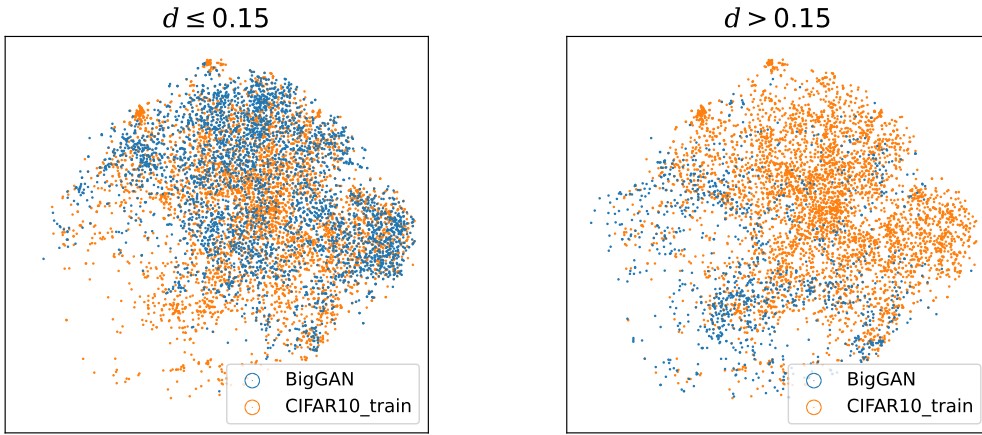

Figure 12: tSNE plot of CIFAR10 "car" 50k generated samples from BigGAN split with $d = 0.15$. The left with generated samples of $d \le 0.15$ covers most of the data distribution while generated samples in the right figure with $d > 0.15$ lies outside of the distribution.

## 6 Conclusion

Training sample memorization is a known issue for GANs but is often addressed as a caution only after models are trained. To the best of our knowledge, we are the first to directly tackle the issue of memorization reduction during GAN training. Specifically, we proposed a training strategy to reject memorized samples when updating the generator. We showed through experiments that our method is effective at reducing memorization and the rejection threshold serves as a control knob for tuning the magnitude of regularization. Selecting a good rejection threshold allows the model to learn to generate from the training distribution but not replicating near duplicates of training samples.

Currently our method discards update information provided by memorized samples to reduce memorization. The information could potentially be better utilized in other ways to further improve generation quality. We hope that the foundation we established inspires future works to explore active memorization reduction techniques for GANs, improving the Pareto frontier of reduced memorization and better generation quality.

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
