# OpenReview forum: "Reducing Training Sample Memorization in GANs by Training with Memorization Rejection"
_TMLR — Rejected by TMLR_

### Review · Reviewer_zQtu · 2022-09-02

**Summary Of Contributions:**

The paper tries to analyze the sample memorization issue in GANs and proposes a simple trick called memorization rejection to solve it. Some empirical results are insightful but the whole paper lacks novelty and has limited impact.


**Requested Changes:**

Other detailed comments are listed as follows:

- Regarding the argument that "the nearest neighbor distance is not an indicator of quality when the distance is sufficiently small": It seems that the only evidence is the qualitative results in Figure 3. Do you cherry-pick the images? I admit that these rows have imperceptible differences, but are they really close enough to each other quantitatively (e.g., measured by FID/IS)?

- "However, in practice performing MR only when training the generator is sufficient": Is this because GAN's discriminator is easy to converge, so you need to include those hard fake samples (which are very close to the training data) to avoid trivial convergence of the discriminator? BTW, do you have an ablation study on this to strengthen this part of the paper?

- The last paragraph of Sec 3.1 may be an algorithm but it collapses

- The computational complexity is high in large-scale cases. On one hand, the method needs to store the embeddings of all training instances; on the other hand, at each training iteration, the method has to do hundreds of times (i.e. the batch size) exact/approximate nearest neighbor search over potentially millions of candidates. Both of these are unacceptable when processing modern datasets like ImageNet and beyond.

- In Figure 6, is the result of $\tau=0.13$ an average value or from one single trial? It is very strange to see that it leads to better FID than $\tau<0.13$. Why?

- Comparisons to SOTA GAN models in aspects of both visual quality and memorization severity are necessary. Large-scale evaluation is also desired as CIFAR-10 data is less diverse than popular real-world image datasets.

- It seems that different GAN models correspond to different C_T vs. FID plots (Figure 9). Do they prefer the same threshold $\tau$? Namely, does the proposed method require intensive hyper-parameter tuning when facing different GAN methods/datasets? Considering Figure 9, Sec 5.7 cannot *generally* solve the problem of the specification of $\tau$

**Strengths And Weaknesses:**

Overall, this is an incremental paper with limited novelty and insufficient empirical studies.

Basically, the memorization rejection is to early stop the training of the generator on some well-trained samples $z$. As agreed by the authors, the early-stopping trick has been investigated by the GAN community, so the technical novelty of the paper is poor.

Besides, I worry that the proposed method has poor scalability and perhaps relies on intensive hyperparameter tuning in practice. Moreover, the authors conducted experiments on only the CIFAR-10 dataset. I want to know if the memorization issue still exists for large-scale datasets like ImageNet and if the proposal still works.

---

### Review · Reviewer_o82S · 2022-09-02

**Summary Of Contributions:**

The paper addresses the problem of sample memorization in GANs. The paper observes that for the generated images with a sufficiently small distance to the nearest training image, further reducing the distance would result in memorization but does not improve the sample quality too much. Motivated by this observation, the paper proposes to discard generated images whose distance to the nearest training image is within a threshold during training. It can be seen as a regularization that penalizes the generator for copying the training images. Experiments show that the proposed approach can reduce memorization without impacting sample quality scores too much across several GAN architectures on the CIFAR10 dataset.

**Broader Impact Concerns:**


I don't have concerns about this.


**Requested Changes:**


[Experiments]
* [Critical] The abstract claims "Experiments on multiple datasets", but there are only experiments on the CIFAR10 dataset. It is important to have experiments on other datasets for supporting the claims and demonstrating the value of the work.
* [Less critical] It would be better to have an ablation study of performing MR only on the generator v.s. performing MR on both the generator and the discriminator.

[Writing]
* [Less critical] Section 2: "X_T\subseteq \mathcal{X}" -> "X_T\subseteq \mathcal{X}^N"
* [Critical] Figure 3: Visual inspection of images is too subjective. For me, I would say that the last row is indeed visually worse than the first row, as opposed to "there is no perceptible quality difference between rows" claimed in the paper. To support the claim, it is better to have a quantitative evaluation besides this qualitative visualization.
* [Less critical] Figure 4: are you doing tSNE from the pixel space or an embedding space? It is not stated in the paper.
* [Less critical] Section 3.1: Compared with the vanilla GAN, this approach has additional undesired equilibriums, where the generator mode-collapse to any single training image, and the discriminator outputs a constant no matter what the input is. I understand that the modified version where MR is only performed on the generator would eliminate this equilibrium, but there could be others. It is probably good to have discussions around this in the paper.
* [Critical] The format of algorithm 3.1 is broken.
* [Critical] Section 3.2 mentioned that an embedding space is used for computing the nearest distance. Which embedding space is used? This should be mentioned in the paper.
* [Less critical] Section 5.2: figure -> Figure
* [Less critical] Figure 5: which tau value is used for BigGAN+MR?
* [Critical] Section 5.4 states "Figure 9 shows that BigGAN with differential augmentation exhibits the most severe memorization when no MR is applied." But in Figure 9, it is unclear which points are the ones without MR.
* [Critical] Figure 8: In the previous texts, it was mentioned that negative C_T means memorization, and a zero C_T means no memorization. What is the meaning of a positive C_T then?
* [Critical] A related question to the above. I don't quite get the discussion in Section 5.6 "The absolute C_T values of the testing ... and C_T value holds." Would you mind explaining it in the rebuttal? Thank you!
* [Less critical] Section 5.6: The dataset names used here are different from what was used in Section 5.1.2. For example, Barz & Denzler (2020) is marked with CIFAIR10 in 5.6, but ciFAIR10 in 5.1.2.
* [Critical] Section 5.7: I understand Figure 12. But why does that imply that "the average nearest neighbor distance of the training samples can be used to determine whether a generated sample is close enough"?

**Strengths And Weaknesses:**


Strengths:
* The idea is simple and intuitive. It is compatible with a wide range of GAN architectures and formulations.
* The results on the CIFAR10 dataset look promising.

Weakness:
* The paper needs a careful pass on the writing. There are many typos and writing issues.
* Although the abstract claims "Experiments on multiple datasets", I only see experiments on the CIFAR10 dataset. Experiments on a single, low-resolution dataset are insufficient to demonstrate the value of the work. It is unclear whether the claimed benefits generalize to other more realistic datasets.

---

### Review · Reviewer_k4Sb · 2022-09-04

**Summary Of Contributions:**

This paper tries to solve the sample memorization issue of deep generative models by first detecting if a generated sample is sufficiently close to the training set and then use early stopping if this is the case. Experimental results on CIFAR-10 datasets are provided to empirically justify the claims.

**Broader Impact Concerns:**

No concerns on the ethical implications of the work.

**Requested Changes:**

Please see the "Strengths And Weaknesses" section for questions/comments.

**Strengths And Weaknesses:**

The paper tries to tackle a well-known issue to generative model - sample memorization. It is true that most popular evaluation metrics such as FID or inception score cannot effectively detect if a generative model has the sample memorization issue and indeed in many application scenarios such as drug/material discovery and art, generative novel samples instead of simply memorizing training set is important. So I agree this is an important question to study.

The main methodological contribution is to detect if a generated sample is sufficiently close to training set in terms of some distance metric (e.g. an embedding function + cosine similarity) and then perform early stopping on those samples. Based on my understanding, both the memorization detection method and the idea of early stopping have been studied previously, and from my point of view, it seems the combination of them does not provide enough theoretical insights/novelty. Below are some detailed questions/comments:

- How to efficiently detect if the model is memorizing the training data is perhaps the first and most important question to consider in this setting. For example, we definitely should not measure L2-distance in pixel space as shifting an image by one pixel is almost just memorizing the image despite a large L2-distance. So employing a feature extraction and measuring similarity in latent space is necessary. But is using a standard architecture like inception-v3 is good enough? Most modern deep NNs have adversarial examples. If we add a negligible noise to an image (the resulting image looks almost the same as before, thus memorization), in the latent space, the new image may be quite different (classified as a different class). So we should be more careful about this part and to really tackle this problem, we may need additional technical consideration instead of plug-and-play.
- "the nearest neighbor distance is not an indicator of quality when the distance is sufficiently small": this kind of argument is more like an intuition rather than a rigorous theoretical statement. For example, how would you prove this (under what assumptions over the function f and how to define quality mathematically)?
- Is there any theoretical guarantee (convergence, unbiasedness etc) for the new objective function (some kind of adaptive importance sampling with dynamically changing proposal distribution), which corresponds to some **modified version** of a well-defined divergence function like f-divergence?
- if the f is the inception network later used in evaluation (e.g. FID), then this may be a leak of information from test to train stage, and the model may have the chance to overfit/exploit to this evaluation metric (the comparison may not be fair for other models).
- At the end of Section 3.1, the algorithm is hard to parse in its current form.
- In the experiments, only CIFAR-10 is used, which cannot really justify the arguments about scaling up to larger datasets, as the computation still seems quite expensive. Also memorization issue seems to be a common problem shared by all deep generative models. So it would be more convincing to include more datasets and more scenarios such as other SOTA GANs and generative models like VAE, diffusion model/EBMs, etc.

---

### Author Response · Authors · 2022-09-18
**Response to key points in reviews**

We sincerely thank the reviewers for the detailed feedback. Two key points shared in multiple reviews are issued in individual sections below:
### Memorization rejection is not just early stopping
Early stopping when the evaluation metric (e.g. FID) starts to deteriorate is indeed a widely-adopted technique for training GANs. The wide adoption is based on the belief that the advasarial optimization process is not guaranteed to converge to an equilibrium where the evaluation metric is optimized. Thus, the purpose of early stopping is to handle to misalignment between the objective function and the ultimate goal (good generation quality), but not necessarily to deal with the issue of training sample memorization.

Our proposed method performs sample rejection during training, which could be thought of as a form of adaptive early stopping. On one hand, this sheds light as to why general early stopping may be effective: to prevent memorization of training sample to the point of deteriorating the evalution metric. On the other hand, we have also shown that early stopping when the evaluation metric starts to deteriorate is not remotely sufficient to prevent training sample memorization (evaluated with the $C_T$ metric). We have shown that standard well-trained GANs that performs well in terms of FID, even when early stopped, still exhibits strong memorization. On the flipside, our proposed memorization rejection technique effectively prevents training sample memorization. It is true that memorization rejection, in terms of the technical implementation, is a form of early stopping based on memorization distance. However, memorization rejection is a solution to the training sample memorization problem, whereas early stopping is not. Novelty is a subjective matter of personal taste. In terms of insight, we revealed how instance-wise early stopping serves as a form of regularization by decoupling the memorization rejection GAN objective.

### Limited experiments on multiple datasets
We completely agree that experimenting only on CIFAR10 is not ideal. It is true that most research involving image generation quality of GAN at least experiments on ImageNet. Our dilemma is the lack of existing rigorous study on "clean" holdout testing samples on other datasets, unlike the meticulously curated CIFAR10.1. Training sample memorization can only be accurated estimated if baseline can be established via comparing with clean testing sets. It pains us that so far to the extend of our knowledge, CIFAR10 is the only dataset with such clean variants of holdout testing set available. It would be immensely beneficial if the reviewers know of any other datasets with rigourously examined holdout clean testing sets to share with us.

Scalability with respect to larger datasets is not an issue if we replace the exact nearest neighbor search (NNS) with approximated nearest neighbor search (ANNS). Recently development in ANNS shows amazing performance, requiring a fraction of runtime to accurately estimate the nearest neighbor distance. As our proposed method does not rely on exactly recovering the nearest neighbor, ANNS serves as a valid drop-in replacement for NNS whenever the rejection steps becomes a bottleneck during training.

---

### Decision · Action_Editors · 2022-10-07

**Recommendation:** Reject

**Comment:**

The three initial reviews raised significant doubts regarding the evidence to support of the claims and also asked clarification questions. One of the reviews also discussed novelty. As novelty (and significance) are not part of TMLR's evaluation criteria, I did not take these comments into account. The authors then provided a response that addressed the most significant criticisms raised by reviewers. The reviewers' evaluation did not change in light of these responses, and all suggest that the paper is not ready for publication at TMLR.

Below I summarize the two main aspects that had the most influence on the decision:

+ A Single dataset is studied. The three reviewers found that it was difficult to support the claim about the method's efficacy since it was validated on a single dataset (as a minor point: the abstract claims it is validated on multiple datasets). The size of that dataset was also discussed, but this is a secondary concern.

  In their reply, the authors state that to obtain accurate memorization results a reference test set disjoint from the training data must be available. This is the case for CIFAR10.1---a "cleaned-up" version of the original CIFAR10 test set---but test sets from large-scale object recognition datasets likely contain near-duplicates from the training set.

  While one reviewer explicitly recognized this reference-set challenge, the reviewers all seem to believe that it would still be valuable to study existing larger-scale datasets. That is, *relative* improvements in memorization still seem meaningful even though an additional level of analysis might be required.

  In addition, setting aside size, studying synthetic datasets might be able to at least show that the method generalizes beyond CIFAR10. It was also suggested that another possibility would be to show that the proposed method generalizes to other deep generative models.

+ The claim that "the nearest neighbor distance is not an indicator of quality when the distance is sufficiently small" was also questioned by reviewers. This is a core assumption for the development of the method. While some qualitative evidence is provided in support (Figure 3), it would be worth confirming this assumption with either further empirical evidence, ideally including quantitative evidence, and/or a theoretical development.

There were a few more minor comments that I suggest the authors should address:
+ Scalability. In their response, the authors say that their method can use approximate neighbor search and so should scale to larger datasets. I would suggest adding a short discussion in the paper about this and ideally verifying it empirically.
+ Writing quality. Two reviewers also suggest that the quality of the presentation could be improved (e.g. the presentation of Algorithm 3.1, see details in the reviews) and further proof-reading the paper was suggested.
+ Unanswered questions. The authors did answer two of the main concerns raised by reviewers in their response, but the reviewers also asked several other clarification questions which seemed not to have been answered at this stage.

I would be willing to consider a significantly revised version of the manuscript.

**Audience:**

Yes. This paper proposes a method for preventing memorization while preserving sample quality in generative models trained in the GAN framework. Ideas involving generating models and improving their training are of interest to the TMLR audience at large. Further, as evidenced by some of the references in the manuscript, the topic of memorization in generative models as well as the proposed method would also be of interest to the TMLR audience.

**Claims And Evidence:**

The evidence provided in support of the claims is accurate (modulo some clarification questions raised by the reviewers) and clear.

However, the empirical study only uses a single dataset. The three reviewers agree that the study does not convincingly validate the claim that the proposed method (memorization rejection) "effectively reduces training sample memorization" (from the abstract).

As mentioned above, the reviewers also asked several clarification questions which should be addressed by the authors. These clarifications would validate the accuracy of some of the claims.